# Stabilization of interdependent networks with two sub-networks of non-identical nodes

Cao Chen, Changyuan Guo[iD]*, Tong Wang

School of Computer Science and Engineering, Chongqing Three Gorges University, Chongqing, China

* guocyup@gmail.com

## Abstract

This paper explores the stabilization of interdependent networks comprising two sub-networks with non-identical nodes, in which one of the sub-networks is connected to the other in one-to-many mode. Firstly, we establish a mathematical model where the sub-networks possess different number of nodes. Besides, the outer coupling matrix is not required to satisfy the diffusive coupling condition. Then, based on some useful assumptions, adaptive decentralized controllers are designed to realize asymptotic stabilization of the system, the validity of the proposed controllers is rigorously established using Lyapunov stability methods. Finally, their effectiveness is demonstrated through two simulation examples.

## Introduction

Practical systems with a number of mutually coupled entities can be modeled as complex networks, as a result, research on complex networks has attracted widespread attention, leading to significant advancements over the past decades [1–3]. Actually, many real systems are composed of several interdependent sub-systems, such as transportation systems consisting of buses, taxis and subways [4], urban infrastructure consisting of power supply system, communication system and transportation system [5,6]. That is to say, the complex network representing such a system consists of several sub-networks. It is important to point out that there is no universally accepted terminology to describe such networks. One can find terms like networks of networks [7], interdependent networks [8,9], multi-layer networks [10,11] or multiplex networks [12] in the existing literature. So, to eliminate ambiguity, we use the term *interdependent networks* in this paper.

In recent years, a growing number of results on interdependent networks have been reported in the literature [13–32], such as cascading failures [15], percolation [19], robustness analysis [23,30], stability [22], and spreading process [25,26]. Synchronization, as an important dynamical property, has been a hot topic of complex networks over the years [33]. Intra and inter-layer synchronization of interdependent networks was studied in Ref. [16]. Gao et al. [17] investigated the synchronization problem of interdependent networks where the links between sub-networks are

**Data availability statement:** All relevant data are within the manuscript and its Supporting information files. Besides, these data are available at the public repository: https://github.com/Chencao0928/Stabilization-of-Interdependent-Networks.

**Funding:** This work was supported by the Scientific and Technological Research Program of Chongqing Municipal Education Commission (KJQN202301218 to CC;KJQN202301219, KJQN202301245 and KJQN202301231 to CG; KJQN202301215 to TW) and the Foundation of Intelligent Ecotourism Subject Group of Chongqing Three Gorges University (zhlv-20221024 to CC).

**Competing interests:** No authors have competing interests.

unidirectional. In Ref. [24], the authors solved complete synchronization problem of interdependent networks. Refs. [21,22] studied inter-structure impacts on the synchronizability and stability of interdependent networks respectively. Li et al. [18] presented an aperiodically adaptive intermittent pinning controller to stabilize the interdependent networks with noise-based superior couplings.

To the best of our knowledge, the stabilization of interdependent networks has received limited attention. Stability refers to the ability of all nodes within a network to return to equilibrium from any arbitrary state solely based on the network's inherent properties. It is a fundamental prerequisite for engineering applications. In general, it is hard for complex networks to reach stable state on their own, let alone interdependent networks with more complicated structures. Therefore, designing effective controllers is essential to ensure the stabilization of interdependent networks. Many effective control schemes have been proposed [34–36], among which the decentralized controller offers advantages in easier implementation and lower dimensionality [37] . Due to these merits, this methodology has been applied widely in large-scale systems. For example, in Refs. [17,35,38,39], decentralized controllers are designed to solve problems in complex networks and interdependent networks.

One common assumption adopted by most of the previous works [16,24,28,32] is that the two sub-networks contain the same number of nodes, and in particular, the studies [17,24,28,32] further assume that all nodes in each sub-network follow identical dynamics. However, such assumptions are not always consistent with reality. In many real-world systems, the subsystems are heterogeneous. For instance, vehicle platoon systems usually consist of vehicles with heterogeneous dynamics, which, as shown in [40,41], can be readily verified to satisfy quadratic (QUAD) conditions. Motivated by these practical settings, this work focuses on the stabilization of heterogeneous interdependent networks without imposing restrictions on identical node numbers or homogeneous dynamics. Removing these limitations fundamentally changes the dynamics of interdependent networks and may cause the failure of most previously proposed control strategies.

Building on the above analysis, this paper explores the stabilization of interdependent networks, where each non-diffusively coupled sub-network consists of non-identical nodes. The main contributions of this work are as follows:

1. The mathematical model of interdependent networks with two sub-networks is proposed. In contrast to previous works [17,24,28,32], where the sub-networks are assumed to contain the same number of identical nodes, our model allows the two sub-networks to have different numbers of nodes as well as different node dynamics. Moreover, the commonly imposed diffusive coupling condition on the outer coupling matrix is not required in this paper.

2. An adaptive decentralized control scheme is proposed to achieve asymptotic stabilization of the aforementioned model, and this scheme is also valid even if the sub-network has identical nodes or the number of nodes in it is the same. The feasibility of the controllers is verified by theoretical results and two numerical simulation examples.

The remainder of this paper is structured as follows: Section 2 presents the necessary assumptions, key lemmas, and the mathematical model of interdependent networks. The control schemes and the proof of theoretical results are presented in main result section. Two simulation examples are presented to verify the effectiveness of the controllers in simulation example section.

## Preliminaries and model formulations

Consider an interdependent network made up of two sub-networks $X$ and $Y$, where $Y$ depends on $X$ in one-to-many mode [42] unidirectionally in Fig 1. Then, the equations of two sub-networks can be described as follows:

$$\dot{x}_i = f_i(x_i) + \sum_{j=1}^{N_x} a_{ij} H_x(x_i) x_j \tag{1}$$

$$\dot{y}_i = g_i(y_i) + \sum_{j=1}^{N_y} b_{ij} H_y(y_i) y_j + \sum_{j=1}^{N_x} c_{ij} H_{xy}(y_i) x_j \tag{2}$$

where, for sub-network $X$, $x_i \in \mathbb{R}^n$ is the state vector; $f_i(x_i) : \mathbb{R}^n \to \mathbb{R}^n$ is the function vector; $H_x(x_i) : \mathbb{R}^n \to \mathbb{R}^n$ is the inner coupling function matrix; $N_x$ represents the number of nodes in $X$; $A = [a_{ij}] \in \mathbb{R}^{n \times n}$ is the outer coupling matrix in which $a_{ij} = a_{ji} > 0$ if there exists a connection from node $j$ to node $i$, otherwise $a_{ij} = 0$; $C = [c_{ij}]$ is the outer coupling matrix between sub-networks $X$ and $Y$, in which, $c_{ij} > 0$ if there exists a dependency from node $j$ in sub-network $X$ to node $i$ in sub-network $Y$, otherwise $c_{ij} = 0$; $H_{xy}(y_i) : \mathbb{R}^n \to \mathbb{R}^n$ is the inner coupling function matrix between $X$ and $Y$, and $y_i, g_i(y_i), H_y(y_i), N_y, B = [b_{ij}]$ in $Y$ are the same as $x_i, f_i(x_i), H_x(x_i), A$ in $X$.

**Remark 1**. Without loss of generality, let $x_i^* = y_i^* = O_n$ be the equilibrium point, i.e. $f_i(x_i^*) = g_i(y_i^*) = O_n$, where $O_n$ is a zero vector.

**Remark 2**. $N_x$ and $N_y$ in Eqs (1)–(2) can be different, which implies that the number of nodes in each sub-network does not have to be the same. This is consistent with reality.

**Remark 3**. If the row sum of an outer coupling matrix equals zero, then it satisfies the diffusive coupling condition [43]. Here, the outer coupling matrices of network model (1)–(2) are not restricted by this condition.

**Assumption 1**. For any given state vectors $x_i \in \mathbb{R}^n$ ($i = 1, \ldots, N_x$) and $y_i \in \mathbb{R}^n$ ($i = 1, \ldots, N_y$), there exists a positive constant $m > 0$ such that $|H_x(x_i)| \leq m$, $|H_y(y_i)| \leq m$, and $|H_{xy}(y_i)| \leq m$.

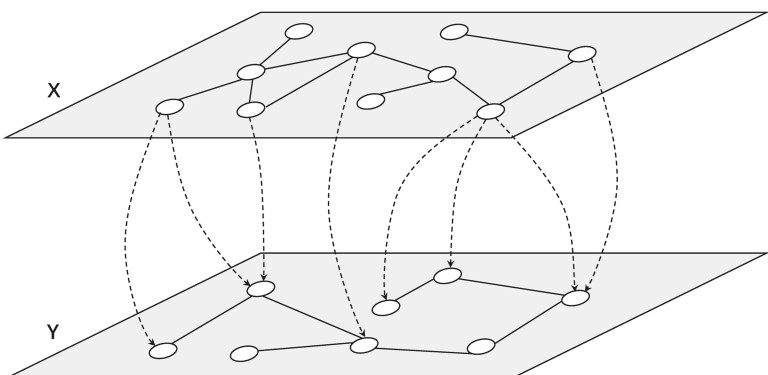

**Fig 1**. A one-to-many interdependent network with two sub-networks $X$ and $Y$.

**Assumption 2**. For any given state vector $\xi_1, \xi_2$ and any $f_i(\cdot)$ there exists a positive constant $\gamma_i \in R$ such that

$$(\xi_1 - \xi_2)^T (f_i(\xi_1) - f_i(\xi_2)) \leq \gamma_i (\xi_1 - \xi_2)^T (\xi_1 - \xi_2). \tag{3}$$

**Assumption 3**. For any given state vector $\xi_1, \xi_2$ and any $g_i(\cdot)$ there exists a positive constant $\beta_i \in R$ such that

$$(\xi_1 - \xi_2)^T (g_i(\xi_1) - g_i(\xi_2)) \leq \beta_i (\xi_1 - \xi_2)^T (\xi_1 - \xi_2). \tag{4}$$

  **Remark 4**. Assumptions 2 and 3 are derived from the QUAD (Quadratic) condition, which generalizes the Lipschitz property for nonlinear systems [44]. These assumptions ensure that the growth rate of the nonlinearities of both sub-networks are quadratically bounded. Since many real-world systems (e.g., power systems and chaotic systems) naturally satisfy QUAD conditions [28,45], these assumptions are practically reasonable.
  **Lemma 1** [46]. If matrix $D > 0$, $D \in R^{n \times n}$ for any given $x, y \in R^n$, then the following inequality holds:

$$2x^T y \leq x^T D x + y^T D^{-1} y \tag{5}$$

  To achieve asymptotic stabilization of the system, adaptive decentralized controllers $u_i^x$ and $u_i^y$ are added to sub-networks $X$ and $Y$ respectively. Then, the system equations are given by

$$\dot{x}_i = f_i(x_i) + \sum_{j=1}^{N_x} a_{ij} H_x(x_i) x_j + u_i^x \tag{6}$$

$$\dot{y}_i = g_i(y_i) + \sum_{j=1}^{N_y} b_{ij} H_y(y_i) y_j + \sum_{j=1}^{N_x} c_{ij} H_{xy}(y_i) x_j + u_i^y \tag{7}$$

and

$$u_i^x = -d_i^x x_i \tag{8}$$
$$u_i^y = -d_i^y y_i \tag{9}$$

where the positive adaptive parameters $d_i^x$ and $d_i^y$ are updated according to the following laws:

$$\dot{d}_i^x = k_i^x x_i^T x_i \tag{10}$$
$$\dot{d}_i^y = k_i^y y_i^T y_i \tag{11}$$

where $k_i^x$ and $k_i^y$ are positive constants.

## Main results

**Theorem 1**. Controllers (8)–(9) can force the nodes in system (1)–(2) to the objective equilibrium point $x^*$ and $y^*$ asymptotically with Assumptions 1-3 holding.
  **Proof**. Consider the candidate Lyapunov function below:

$$V(t) = V_1(t) + V_2(t) \tag{12}$$

where

$$V_1(t) = \frac{1}{2}\sum_{i=1}^{N_x} x_i^T x_i + \frac{1}{2}\sum_{i=1}^{N_x} \frac{\left(d_i^x - d^x\right)^2}{k_i^x} \tag{13}$$

$$V_2(t) = \frac{1}{2}\sum_{i=1}^{N_y} y_i^T y_i + \frac{1}{2}\sum_{i=1}^{N_y} \frac{\left(d_i^y - d^y\right)^2}{k_i^y} \tag{14}$$

where $d^x$ and $d^y$ are tunable positive constants.

Calculating the time derivative of $V(t)$ along (6) and (7), we can obtain that:

$$\dot{V}_1(t) = \sum_{i=1}^{N_x} x_i^T \left[ f_i(x_i) + \sum_{j=1}^{N_x} a_{ij} H_x(x_i) x_j - d_i^x x_i \right] + \sum_{i=1}^{N_x} (d_i^x - d^x) x_i^T x_i$$

$$= \sum_{i=1}^{N_x} x_i^T \left[ f_i(x_i) + \sum_{j=1}^{N_x} a_{ij} H_x(x_i) x_j - d^x x_i \right] \tag{15}$$

$$\dot{V}_2(t) = \sum_{i=1}^{N_y} y_i^T \left[ g_i(y_i) + \sum_{j=1}^{N_y} b_{ij} H_y(y_i) y_j + \sum_{j=1}^{N_x} c_{ij} H_{xy}(y_i) x_j - d_i^y y_i \right]$$

$$+ \sum_{i=1}^{N_y} \left( d_i^y - d^y \right) y_i^T y_i$$

$$= \sum_{i=1}^{N_y} y_i^T \left[ g_i(y_i) + \sum_{j=1}^{N_y} b_{ij} H_y(y_i) y_j + \sum_{j=1}^{N_x} c_{ij} H_{xy}(y_i) x_j - d^y y_i \right] \tag{16}$$

According to Remark 1 and Assumptions 2-3, we can get that:

$$(x_i - O_n)^T (f_i(x_i) - f_i(O_n)) \le \gamma_i x_i^T x_i \tag{17}$$

$$(y_i - O_n)^T (g_i(y_i) - g_i(O_n)) \le \beta_i y_i^T y_i \tag{18}$$

According to Lemma 1, if $D$ is a identity matrix, then the following inequality holds:

$$\sum_{i=1}^{N_y}\sum_{j=1}^{N_x} c_{ij} y_i^T H_{xy}(y_i) x_j \le \frac{1}{2}\sum_{i=1}^{N_y}\sum_{j=1}^{N_x} c_{ij} y_i^T H_{xy}(y_i) H_{xy}^T(y_i) y_i$$

$$+ \frac{1}{2}\sum_{i=1}^{N_y}\sum_{j=1}^{N_x} c_{ij} x_j^T x_j \tag{19}$$

Next, denote $a = \max_{1 \le i,j \le N_x} |a_{ij}|$, $b = \max_{1 \le i,j \le N_x} |b_{ij}|$, $c = \max_{1 \le i \le N_x, 1 \le j \le N_y} |c_{ij}|$ and $\delta = \max\{\gamma_1, \gamma_2, ..., \gamma_{N_x}, \beta_1, \beta_2, ..., \beta_{N_y}\}$. Then, from the (17)–(19), we can deduce that:

$$\dot{V}(t) = \sum_{i=1}^{N_x} x^T \left[ f_i(x_i) + \sum_{j=1}^{N_x} a_{ij} H_x(x_i) x_j - d^x x_i \right]$$

$$+ \sum_{i=1}^{N_y} y_i^T \left[ g_i(y_i) + \sum_{j=1}^{N_y} b_{ij} H_y(y_i) + \sum_{j=1}^{N_x} c_{ij} H_{xy}(y_i) x_j - d^y y_i \right]$$

$$\leq \sum_{i=1}^{N_x} (\delta - d^x) x_i^T x_i + \sum_{i=1}^{N_x} \sum_{j=1}^{N_x} a_{ij} x_i^T H_x(x_i) x_j$$

$$+ \sum_{i=1}^{N_y} (\delta - d^y) y_i^T y_i + \sum_{i=1}^{N_y} \sum_{j=1}^{N_y} b_{ij} y_i^T H_y(y_i) y_j$$

$$+ \frac{1}{2} \sum_{i=1}^{N_y} \sum_{j=1}^{N_x} c_{ij} y_i^T H_{xy}(y_i) H_{xy}^T(y_i)(x_i) y_i + \frac{1}{2} \sum_{i=1}^{N_y} \sum_{j=1}^{N_x} c_{ij} x_j^T x_j$$

$$\leq \sum_{i=1}^{N_x} (\delta - d^x) x_i^T x_i + \sum_{i=1}^{N_x} \sum_{j=1}^{N_x} a_{ij} \left\| x_i^T \right\| \left\| H_x(x_i) \right\| \left\| x_j \right\|$$

$$+ \sum_{i=1}^{N_y} (\delta - d^y) y_i^T y_i + \sum_{i=1}^{N_y} \sum_{j=1}^{N_y} b_{ij} \left\| y_i^T \right\| \left\| H_y(y_i) \right\| \left\| y_j \right\|$$

$$+ \frac{1}{2} \sum_{i=1}^{N_y} \sum_{j=1}^{N_x} c_{ij} \| y_i^T \| \| H_{xy}(y_i) \| \| H_{xy}^T(y_i) \| \| y_i \| + \frac{1}{2} \sum_{i=1}^{N_y} \sum_{j=1}^{N_x} c_{ij} x_j^T x_j$$

$$\leq \sum_{i=1}^{N_x} (\delta - d^x) x_i^T x_i + m \sum_{i=1}^{N_x} \sum_{j=1}^{N_x} a_{ij} \frac{x_i^T x_i + x_j^T x_j}{2}$$

$$+ \sum_{i=1}^{N_y} (\delta - d^y) y_i^T y_i + m \sum_{i=1}^{N_y} \sum_{j=1}^{N_y} b_{ij} \frac{y_i^T y_i + y_j^T y_j}{2}$$

$$+ \frac{1}{2} m^2 \sum_{i=1}^{N_y} \sum_{j=1}^{N_x} c_{ij} y_i^T y_i + \frac{1}{2} \sum_{i=1}^{N_y} \sum_{j=1}^{N_x} c_{ij} x_j^T x_j$$

$$\leq \sum_{i=1}^{N_x} (\delta - d^x + m a N_x) x_i^T x_i + \frac{1}{2} \sum_{i=1}^{N_y} \sum_{j=1}^{N_x} c_{ij} x_j^T x_j$$

$$+ \sum_{i=1}^{N_y} (\delta - d^y + m b N_y) y_i^T y_i + \frac{1}{2} m^2 \sum_{i=1}^{N_y} \sum_{j=1}^{N_x} c_{ij} y_i^T y_i$$

$$\leq \sum_{i=1}^{N_x} (\delta - d^x + m a N_x) x_i^T x_i + \frac{1}{2} c N_y \sum_{j=1}^{N_x} x_j^T x_j$$

$$+ \sum_{i=1}^{N_y} (\delta - d^y + m b N_y) y_i^T y_i + \frac{1}{2} c m^2 N_x \sum_{i=1}^{N_y} y_i^T y_i$$

$$= \sum_{i=1}^{N_x} (\delta - d^x + m a N_x + \frac{1}{2} c N_y) x_i^T x_i$$

$$+ \sum_{i=1}^{N_y} (\delta - d^y + m b N_y + \frac{1}{2} m^2 c N_x) y_i^T y_i \tag{20}$$

According to (20), Choosing proper $d^x > \delta + maN_x + \frac{1}{2}cN_y$ and $d^y > \delta + mbN_y + \frac{1}{2}m^2cN_x$, we can get:

$$\dot{V}(t) < 0 \tag{21}$$

Based on Lyapunov stability theory, (21) indicates that the interdependent network (6)–(7) is stabilized asymptotically.

**Remark 5**. From the process of the proof above, Theorem 1 still holds in the following two cases:

1. The nodes in each sub-network are identical.
2. Each sub-network has the same number of nodes, i.e. $N_x = N_y$.

## Simulation example

This section provides two examples to verify the effectiveness of controllers (8)–(9). For each example, the sub-network $X$ is a small-world network while the sub-network $Y$ is a scale free network.

**Example 1**. Consider a sub-network $X$ consisting of 15 nodes and a sub-network $Y$ consisting of 10 nodes as follows:

$$\dot{x}_i = f_i(x_i) = \begin{bmatrix} -10x_{i1} + 10x_{i2} \\ b_i x_{i1} - x_{i2} - x_{i1}x_{i3} \\ x_{i1}x_{i2} - \frac{8}{3}x_{i3} \end{bmatrix} \tag{22}$$

$$\dot{y}_i = g_i(y_i) = \begin{bmatrix} 36(y_{i2} - y_{i1}) \\ q_i y_{i2} - y_{i1}y_{i3} \\ -3y_{i3} + y_{i1}y_{i2} \end{bmatrix} \tag{23}$$

where $b_i = 20 + i$, $q_i = 7 + i$. It should be noted that node $x_i (i = 1, 2, ..., 10)$ is Lorenz system [47], while node $y_i (i = 1, 2, ..., 10)$ is Lü system [48]. As we can see, all nodes in (22)–(23) are non-identical. Other parameters are chosen as follows:

$$H_x(x_i) = \begin{bmatrix} \cos(x_{i1}) & 0 & 0 \\ 0 & \cos(x_{i2}) & 0 \\ 0 & 0 & \cos(x_{i3}) \end{bmatrix}$$

$$H_y(y_i = \begin{bmatrix} \sin(y_{i1}) & 0 & 0 \\ 0 & \sin(y_{i2}) & 0 \\ 0 & 0 & \sin(y_{i3}) \end{bmatrix}$$

$$H_{xy}(y_i) = \begin{bmatrix} \arctan(y_{i1}) & 0 & 0 \\ 0 & \arctan(y_{i2}) & 0 \\ 0 & 0 & \arctan(y_{i3}) \end{bmatrix}$$

$$A = 0.1 \begin{bmatrix}
0 & 2 & 3 & 0 & 0 & 0 & 0 & 0 & 0 & 0 & 0 & 0 & 0 & 4 & 0 \\
2 & 0 & 6 & 5 & 0 & 0 & 0 & 0 & 0 & 0 & 4 & 0 & 0 & 0 & 0 \\
3 & 6 & 0 & 0 & 7 & 8 & 9 & 0 & 0 & 0 & 0 & 0 & 0 & 0 & 1 \\
0 & 5 & 0 & 0 & 1 & 0 & 0 & 0 & 0 & 0 & 5 & 0 & 0 & 6 & 0 \\
0 & 0 & 7 & 1 & 0 & 1 & 6 & 0 & 0 & 0 & 0 & 0 & 0 & 0 & 0 \\
0 & 0 & 8 & 0 & 1 & 0 & 0 & 0 & 0 & 0 & 0 & 0 & 0 & 1 & 0 \\
0 & 0 & 9 & 0 & 6 & 0 & 0 & 3 & 0 & 0 & 0 & 0 & 0 & 0 & 7 \\
0 & 0 & 0 & 0 & 0 & 0 & 3 & 0 & 6 & 4 & 0 & 0 & 0 & 0 & 0 \\
0 & 0 & 0 & 0 & 0 & 0 & 0 & 6 & 0 & 8 & 9 & 0 & 0 & 0 & 0 \\
0 & 0 & 0 & 0 & 0 & 0 & 0 & 4 & 8 & 0 & 7 & 2 & 0 & 0 & 0 \\
0 & 4 & 0 & 5 & 0 & 0 & 0 & 0 & 9 & 7 & 0 & 1 & 0 & 0 & 0 \\
0 & 0 & 0 & 0 & 0 & 0 & 0 & 0 & 0 & 2 & 1 & 0 & 1 & 4 & 1 \\
0 & 0 & 0 & 0 & 0 & 0 & 0 & 0 & 0 & 0 & 0 & 1 & 0 & 1 & 8 \\
4 & 0 & 0 & 6 & 0 & 1 & 0 & 0 & 0 & 0 & 0 & 4 & 1 & 0 & 0 \\
0 & 0 & 1 & 0 & 0 & 0 & 7 & 0 & 0 & 0 & 0 & 1 & 8 & 0 & 0
\end{bmatrix}$$

$$B = \begin{bmatrix}
0 & 6 & 5 & 4 & 3 & 2 & 1 & 2 & 3 & 4 \\
6 & 0 & 3 & 4 & 0 & 1 & 0 & 5 & 0 & 1 \\
5 & 3 & 0 & 0 & 1 & 0 & 0 & 0 & 0 & 0 \\
4 & 4 & 0 & 0 & 0 & 0 & 0 & 0 & 0 & 0 \\
3 & 0 & 1 & 0 & 0 & 0 & 7 & 0 & 0 & 0 \\
2 & 1 & 0 & 0 & 0 & 0 & 0 & 0 & 8 & 0 \\
1 & 0 & 0 & 0 & 7 & 0 & 0 & 0 & 0 & 0 \\
2 & 5 & 0 & 0 & 0 & 0 & 0 & 0 & 0 & 0 \\
3 & 0 & 0 & 0 & 0 & 8 & 0 & 0 & 0 & 0 \\
4 & 1 & 0 & 0 & 0 & 0 & 0 & 0 & 0 & 0
\end{bmatrix}$$

$$C = 0.1 \begin{bmatrix}
0 & 0 & 1 & 0 & 0 & 0 & 0 & 0 & 0 & 1 & 0 & 0 & 0 & 0 \\
0 & 0 & 0 & 0 & 0 & 0 & 0 & 0 & 0 & 1 & 0 & 0 & 0 & 0 & 0 \\
0 & 0 & 0 & 1 & 0 & 0 & 0 & 0 & 0 & 0 & 0 & 0 & 1 & 0 & 0 \\
0 & 0 & 0 & 0 & 0 & 1 & 0 & 0 & 0 & 0 & 0 & 0 & 1 & 0 & 0 \\
0 & 1 & 0 & 0 & 0 & 0 & 1 & 0 & 0 & 0 & 0 & 0 & 0 & 0 & 0 \\
0 & 0 & 0 & 0 & 1 & 0 & 0 & 0 & 0 & 0 & 0 & 0 & 0 & 1 & 0 \\
0 & 0 & 0 & 0 & 0 & 0 & 0 & 0 & 0 & 0 & 0 & 1 & 0 & 0 & 0 \\
1 & 0 & 0 & 0 & 0 & 0 & 0 & 0 & 0 & 0 & 0 & 0 & 0 & 0 & 0 \\
0 & 0 & 0 & 0 & 0 & 0 & 0 & 0 & 1 & 0 & 0 & 0 & 0 & 0 & 0 \\
0 & 0 & 0 & 0 & 0 & 0 & 0 & 1 & 0 & 0 & 0 & 0 & 0 & 0 & 1
\end{bmatrix}$$

The initial values are chosen as $x_i(0) = [i, -i, i]^T$, $d_i^x(0) = 0.1$, $k_i^x = 10$, $i = 1, 2, \cdots, 15$, $y_i(0) = [i, i, -i]^T$, $d_i^y(0) = 0.2$, $k_i^y = 20$, $i = 1, 2, \cdots, 10$, and simulation results are presented in Figs 2–7.

The uncontrolled state responses of the interdependent network (22)–(23) are illustrated in Figs 2–3. Application of controllers (8)–(9) results in rapid stabilization to the equilibrium point, as shown in Figs 4–5. Finally, Figs 6–7 demonstrate that each adaptive compensation $d_i(t)$ converges swiftly to a constant as $t$ approaches infinity.

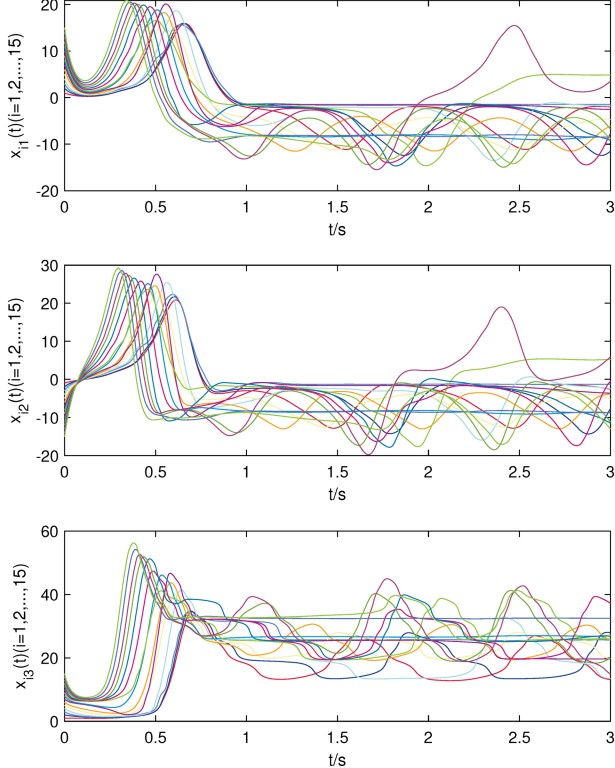

**Fig 2**. The state trajectories of nodes of sub-network X in (22) without controller.

**Example 2**. A different interdependent network is constructed in this example to verify Remark 5. Consider an interdependent network where the sub-network has identical nodes and the number of nodes in it is the same as follows:

$$\dot{x}_i = f_i(x_i) = \begin{bmatrix} -10x_{i1} + 10x_{i2} \\ 28x_{i1} - x_{i2} - x_{i1}x_{i3} \\ x_{i1}x_{i2} - \frac{8}{3}x_{i3} \end{bmatrix} \tag{24}$$

$$\dot{y}_i = g_i(y_i) = \begin{bmatrix} y_{i2} - y_{i3} \\ y_{i1} + 0.1y_{i2} \\ y_{i1}y_{i3} - 14y_{i3} \end{bmatrix} \tag{25}$$

Other parameters are chosen as follows:

$$H_x(x_i) = \begin{bmatrix} \sin(x_{i1}) & 0 & 0 \\ 0 & \sin(x_{i2}) & 0 \\ 0 & 0 & \sin(x_{i3}) \end{bmatrix}$$

$$H_y(y_i) = \begin{bmatrix} \cos(y_{i1}) & 0 & 0 \\ 0 & \cos(y_{i2}) & 0 \\ 0 & 0 & \cos(y_{i3}) \end{bmatrix}$$

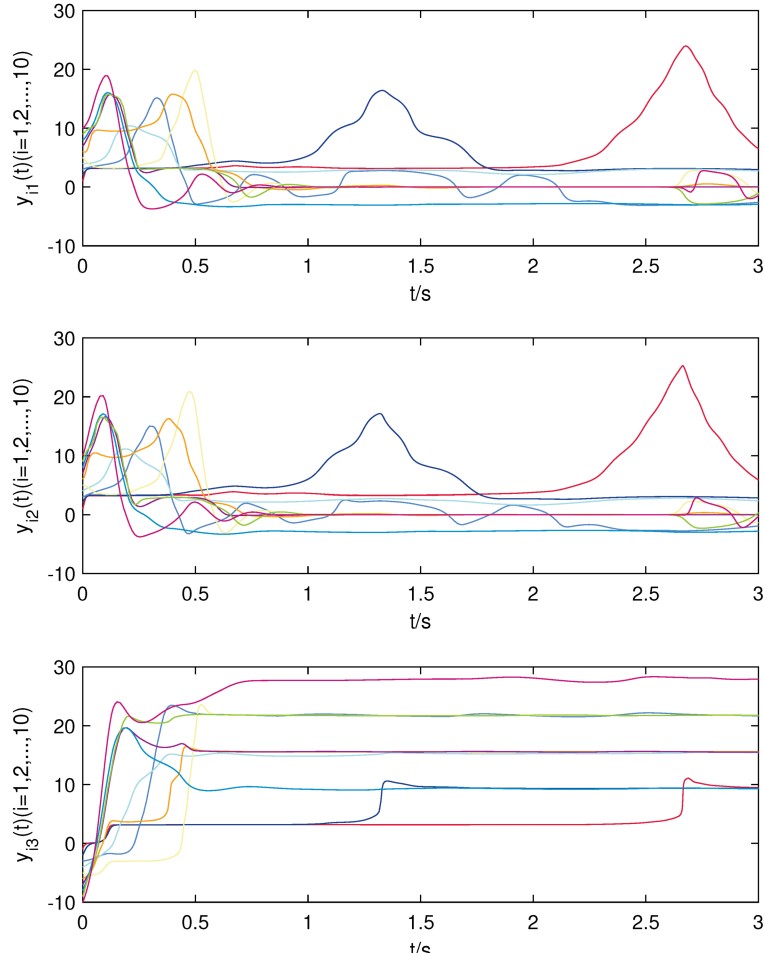

**Fig 3.** The state trajectories of nodes of sub-network Y in (23) without controller.

$$H_{xy}(y_i) = \begin{bmatrix} y_{i1} & 0 & 0 \\ 0 & y_{i2} & 0 \\ 0 & 0 & y_{i3} \end{bmatrix}$$

$$A = 0.1 \begin{bmatrix} 0 & 4 & 3 & 0 & 0 & 0 & 0 & 0 & 5 & 1 \\ 4 & 0 & 0 & 6 & 0 & 0 & 0 & 0 & 2 & 6 \\ 3 & 0 & 0 & 4 & 3 & 2 & 0 & 0 & 0 & 0 \\ 0 & 6 & 4 & 0 & 5 & 1 & 0 & 0 & 0 & 0 \\ 0 & 0 & 3 & 5 & 0 & 0 & 7 & 0 & 0 & 1 \\ 0 & 0 & 2 & 1 & 0 & 0 & 0 & 0 & 4 & 0 \\ 0 & 0 & 0 & 0 & 7 & 0 & 0 & 1 & 2 & 0 \\ 0 & 0 & 0 & 0 & 0 & 0 & 1 & 0 & 1 & 1 \\ 5 & 2 & 0 & 0 & 0 & 4 & 2 & 1 & 0 & 3 \\ 1 & 6 & 0 & 0 & 1 & 0 & 0 & 3 & 1 & 0 \end{bmatrix}$$

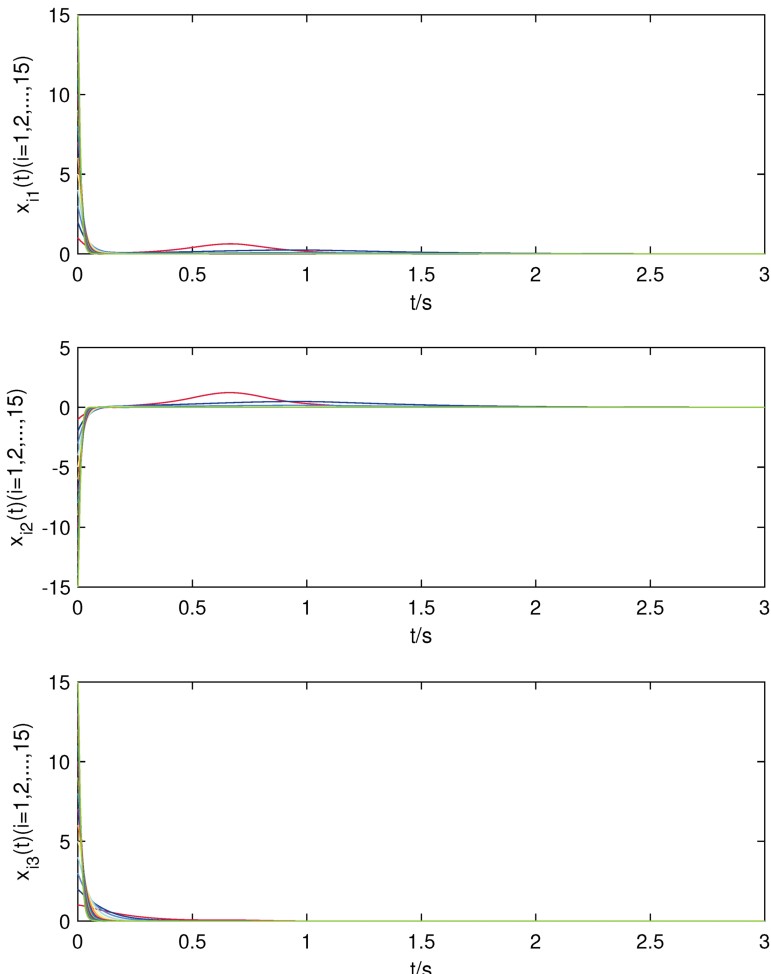

**Fig 4. The state trajectories of nodes of sub-network X in (22) with controller.**

$$B = 0.1 \begin{bmatrix} 0 & 6 & 5 & 4 & 3 & 2 & 1 & 2 & 3 & 4 \\ 6 & 0 & 3 & 4 & 0 & 1 & 0 & 5 & 0 & 1 \\ 5 & 3 & 0 & 0 & 1 & 0 & 0 & 0 & 0 & 0 \\ 4 & 4 & 0 & 0 & 0 & 0 & 0 & 0 & 0 & 0 \\ 3 & 0 & 1 & 0 & 0 & 0 & 7 & 0 & 0 & 0 \\ 2 & 1 & 0 & 0 & 0 & 0 & 0 & 0 & 8 & 0 \\ 1 & 0 & 0 & 0 & 7 & 0 & 0 & 0 & 0 & 0 \\ 2 & 5 & 0 & 0 & 0 & 0 & 0 & 0 & 0 & 0 \\ 3 & 0 & 0 & 0 & 0 & 8 & 0 & 0 & 0 & 0 \\ 4 & 1 & 0 & 0 & 0 & 0 & 0 & 0 & 0 & 0 \end{bmatrix}$$

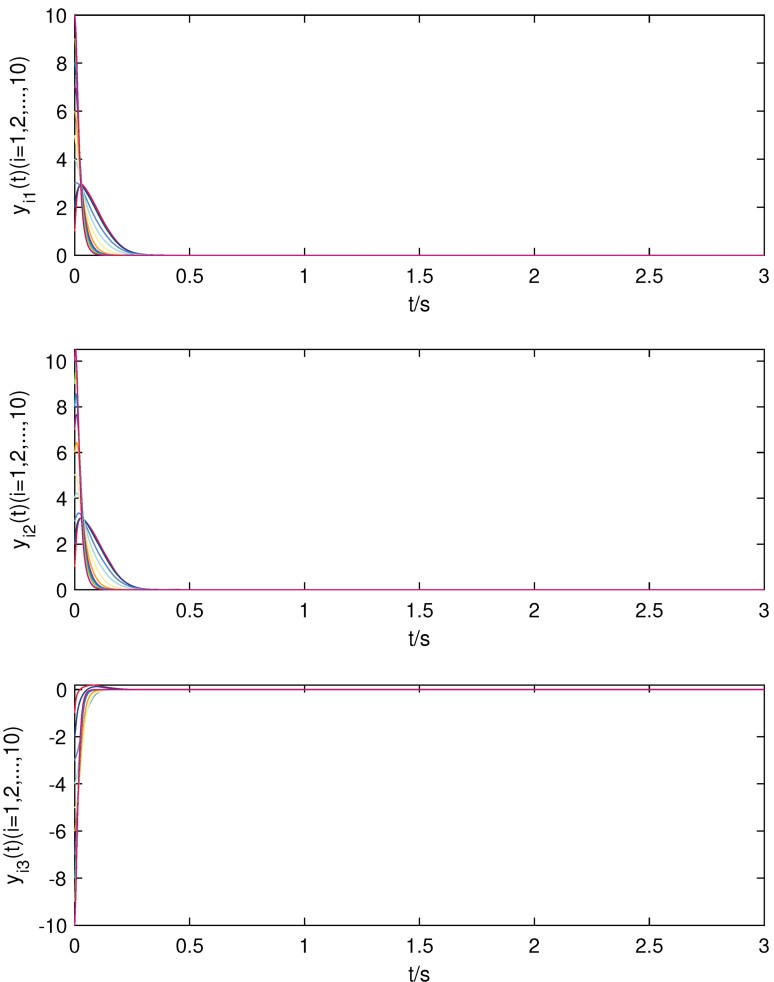

**Fig 5**. The state trajectories of nodes of sub-network Y in (23) with controller.

$$
C = \begin{bmatrix}
0 & 0 & 0 & 0 & 0 & 0 & 0 & 0 & 3 & 0 \\
0 & 0 & 0 & 0 & 0 & 0 & 0 & 0 & 0 & 4 \\
2 & 0 & 0 & 0 & 0 & 0 & 0 & 0 & 0 & 0 \\
0 & 0 & 0 & 1 & 0 & 0 & 0 & 0 & 0 & 0 \\
0 & 5 & 0 & 0 & 0 & 0 & 0 & 0 & 0 & 0 \\
0 & 0 & 9 & 0 & 0 & 0 & 0 & 0 & 0 & 0 \\
0 & 0 & 0 & 0 & 1 & 0 & 0 & 0 & 0 & 0 \\
0 & 0 & 0 & 0 & 0 & 10 & 0 & 0 & 0 & 0 \\
0 & 0 & 0 & 0 & 0 & 0 & 6 & 0 & 0 & 0 \\
0 & 0 & 0 & 0 & 0 & 0 & 0 & 1 & 0 & 0
\end{bmatrix}
$$

The initial states of $X$ are chosen as $x_i(0) = [i, -i, i]^T, d_i^x(0) = 0.1, k_i^x = 10, i = 1, 2, ..., 10$. The initial states of $Y$ are chosen as $y_i(0) = [0.0265 + 0.01i, -0.163 + 0.01i, 0.0602 + 0.01i]^T, d_i^y(0) = 0.2, k_i^y = 10, i = 1, 2, \cdots, 10$. Numerical results are depicted in Figs 8–11.

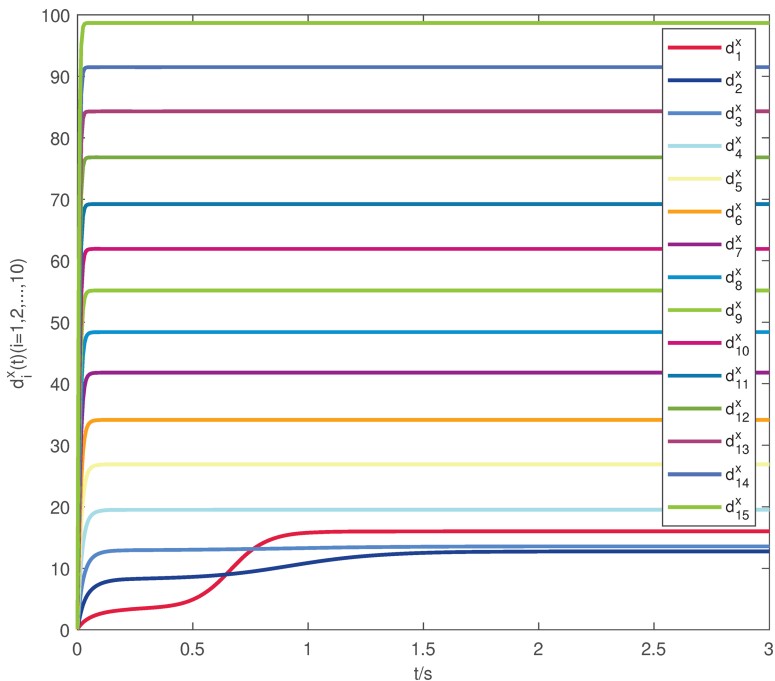

**Fig 6**. The evolution of the $d_i^x(t)$ exerted on system (22)–(23).

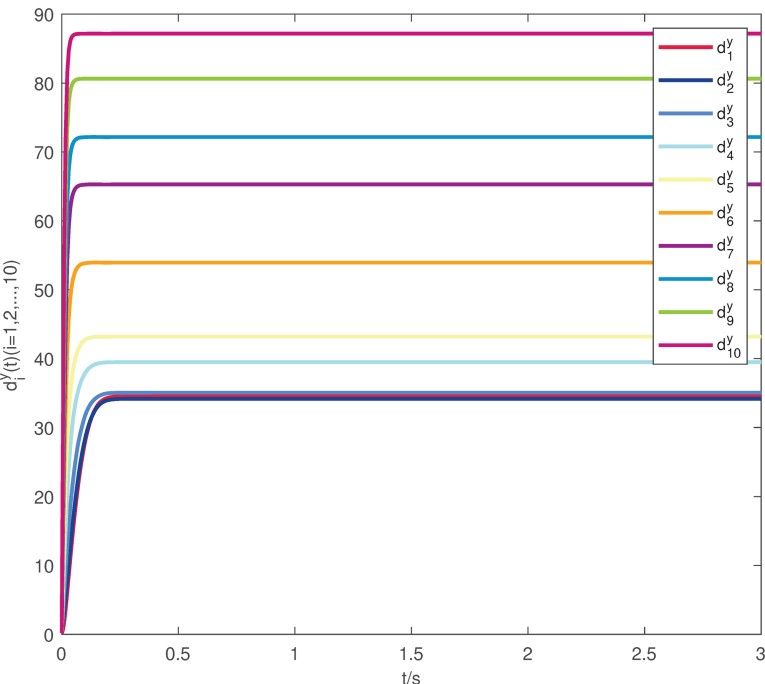

**Fig 7**. The evolution of the $d_i^y(t)$ exerted on system (22)–(23).

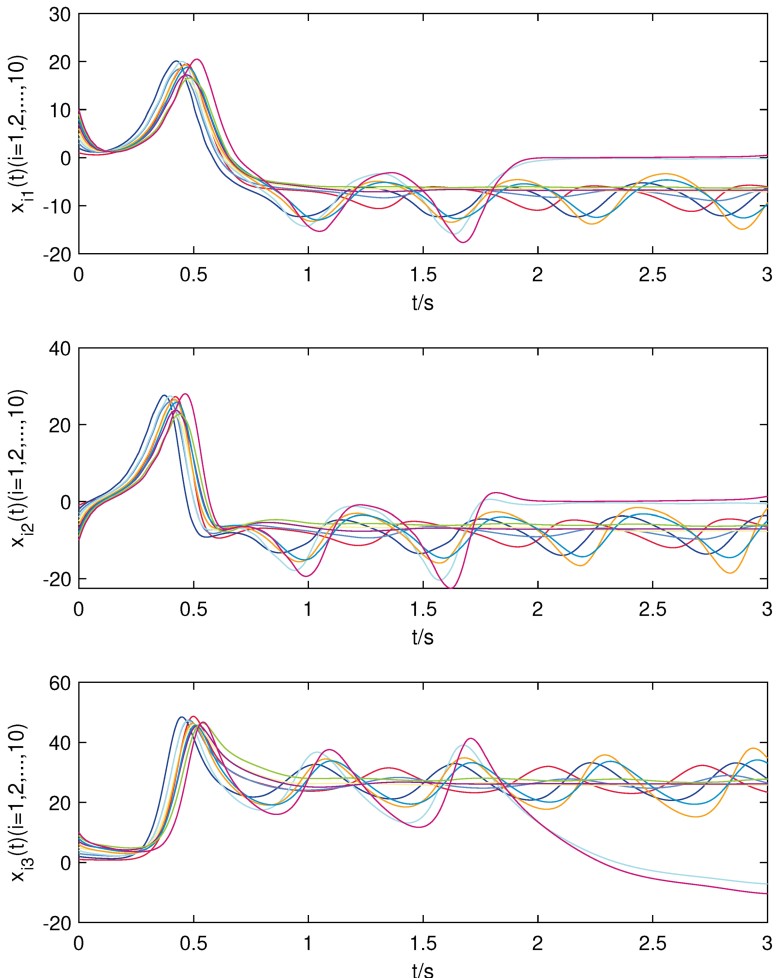

**Fig 8**. **The state trajectories of nodes of sub-network X in (24) without controller.**

Figs 8–9 show that all the uncontrolled nodes in interdependent network (24)–(25) are unstable. From Figs 10–11, it is shown that the stable state of system is achieved quickly which means that controllers (8)–(9) are still valid when the nodes in each sub-network are identical.

**Example 3**. To illustrate the applicability of the proposed stabilization strategy, we consider a task of UAV–UGV (Unmanned Aerial Vehicle–Unmanned Ground Vehicle) formation control. In this example, the UAVs and UGVs are modeled with different nonlinear dynamics, representing a heterogeneous interdependent system. The control objective is to achieve formation rendezvous where both UGVs and UAVs maintain square formations with 2m spacing while moving toward the designated target location $p_d^x = [10, 10]^T$ m for UGVs and $p_d^y = [10, 10, 2]^T$ m for UAVs, with UAVs maintaining a constant height of 2m above the corresponding UGVs. Moreover, the final velocities should converge to zero, i.e., $v_d = 0$, ensuring that the formation reaches a stable equilibrium at the target. Referring to [49], the UAV–UGV group is composed of 4 UGVs (operating in 2D) and 4 UAVs (operating in 3D), and their dynamic models are given as follows:

$$\dot{x}_i = D_{xi} x_i + \sum_{j=1}^{N_x} a_{ij} H_x(x_i) x_j + E_{xi} u_i^x \tag{26}$$

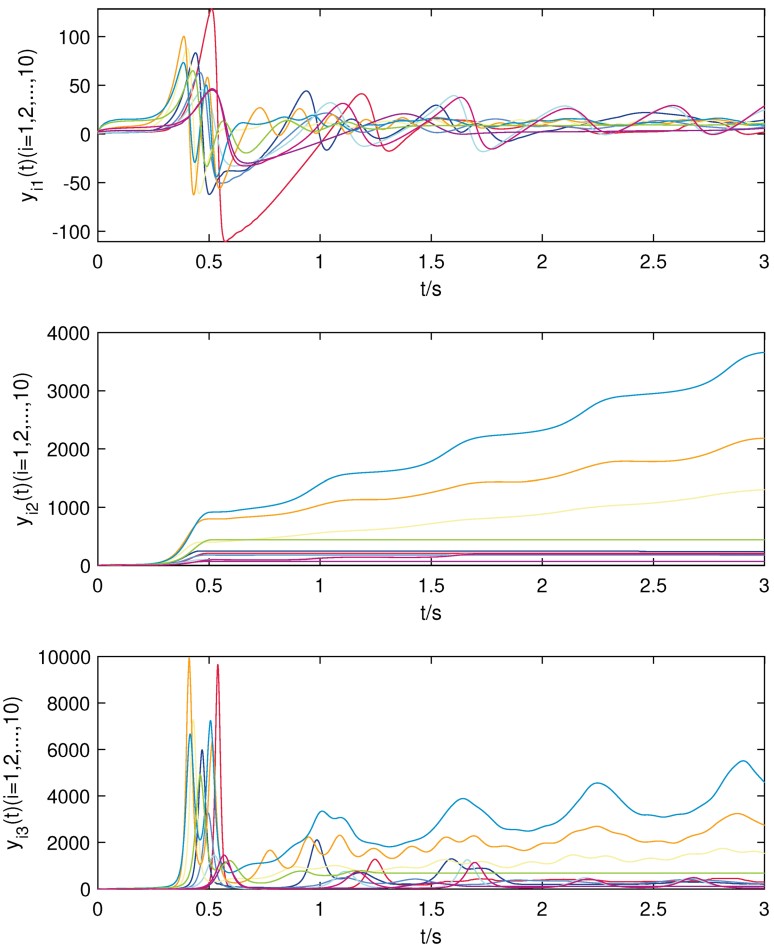

**Fig 9**. **The state trajectories of nodes of sub-network Y in (25) without controller.**

$$\dot{y}_i = D_{yi}y_i + \sum_{j=1}^{N_y} b_{ij}H_y(y_i)\,y_j + \sum_{j=1}^{N_x} c_{ij}H_{xy}(y_i)\,x_j + E_{yi}u_i^y \tag{27}$$

where $x_i \in \mathbb{R}^4$ represents the state vector of the $i$-th UGV containing position and velocity components $[p_{1i}^x, p_{2i}^x, v_{1i}^x, v_{2i}^x]^T$, $y_i \in \mathbb{R}^6$ denotes the state vector of the $i$-th UAV with components $[p_{1i}^y, p_{2i}^y, p_{3i}^y, v_{1i}^y, v_{2i}^y, v_{3i}^y]^T$, $D_{xi} \in \mathbb{R}^{4\times4}$ and $D_{yi} \in \mathbb{R}^{6\times6}$ are the system dynamics matrices for UGV and UAV respectively, $E_{xi} \in \mathbb{R}^{4\times2}$ and $E_{yi} \in \mathbb{R}^{6\times3}$ are the control input matrices.

In simulation, the communication topology matrices are:

$$A = B = \begin{bmatrix} 0 & 1 & 0 & 1 \\ 1 & 0 & 1 & 0 \\ 0 & 1 & 0 & 1 \\ 1 & 0 & 1 & 0 \end{bmatrix}, \quad C = \begin{bmatrix} 1 & 0 & 0 & 0 \\ 0 & 1 & 0 & 0 \\ 0 & 0 & 1 & 0 \\ 0 & 0 & 0 & 1 \end{bmatrix} \tag{28}$$

The system dynamics matrices for UGVs $i$ are:

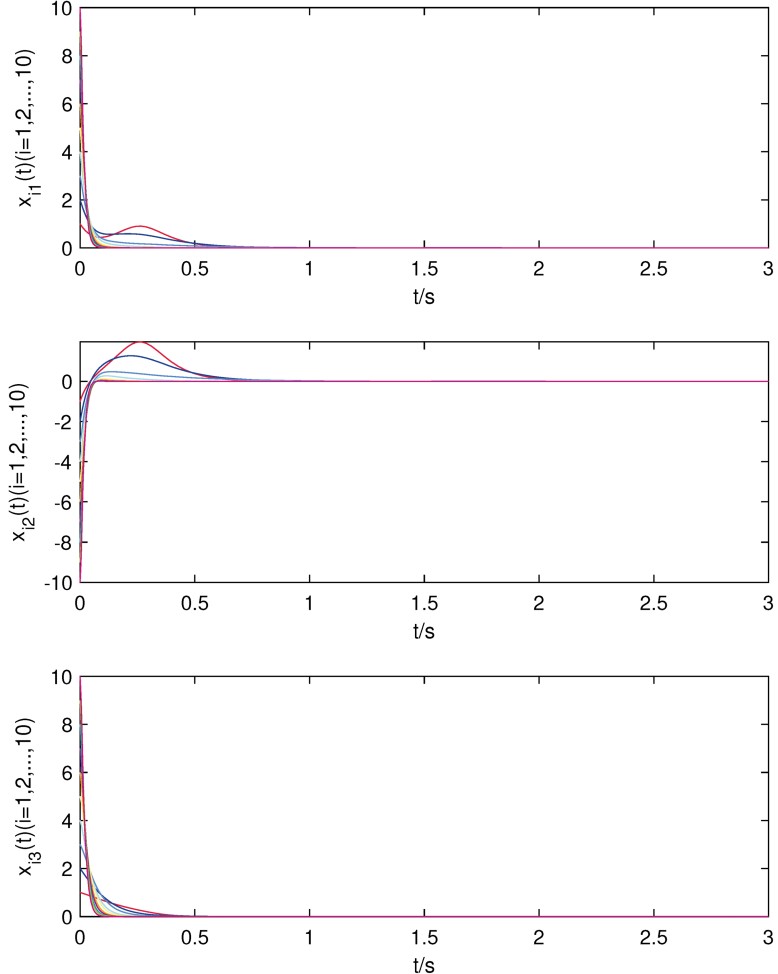

**Fig 10**. **The state trajectories of nodes of sub-network X in (24) with controller.**

$$D_{xi} = \begin{bmatrix} 0 & 0 & 1 & 0 \\ 0 & 0 & 0 & 1 \\ 0 & 0 & -\alpha_i & 0 \\ 0 & 0 & 0 & -\alpha_i \end{bmatrix}, \quad E_{xi} = \begin{bmatrix} 0 & 0 \\ 0 & 0 \\ 1 & 0 \\ 0 & 1 \end{bmatrix} \tag{29}$$

where $\alpha_i = 0.05 + 0.01i$ represents heterogeneous damping coefficients.

The system dynamics matrices for UAV $i$ are:

$$D_{yi} = \begin{bmatrix} 0 & 0 & 0 & 1 & 0 & 0 \\ 0 & 0 & 0 & 0 & 1 & 0 \\ 0 & 0 & 0 & 0 & 0 & 1 \\ 0 & 0 & 0 & -\beta_i & 0 & 0 \\ 0 & 0 & 0 & 0 & -\beta_i & 0 \\ 0 & 0 & 0 & 0 & 0 & -\beta_i \end{bmatrix}, \quad E_{yi} = \begin{bmatrix} 0 & 0 & 0 \\ 0 & 0 & 0 \\ 0 & 0 & 0 \\ 1 & 0 & 0 \\ 0 & 1 & 0 \\ 0 & 0 & 1 \end{bmatrix} \tag{30}$$

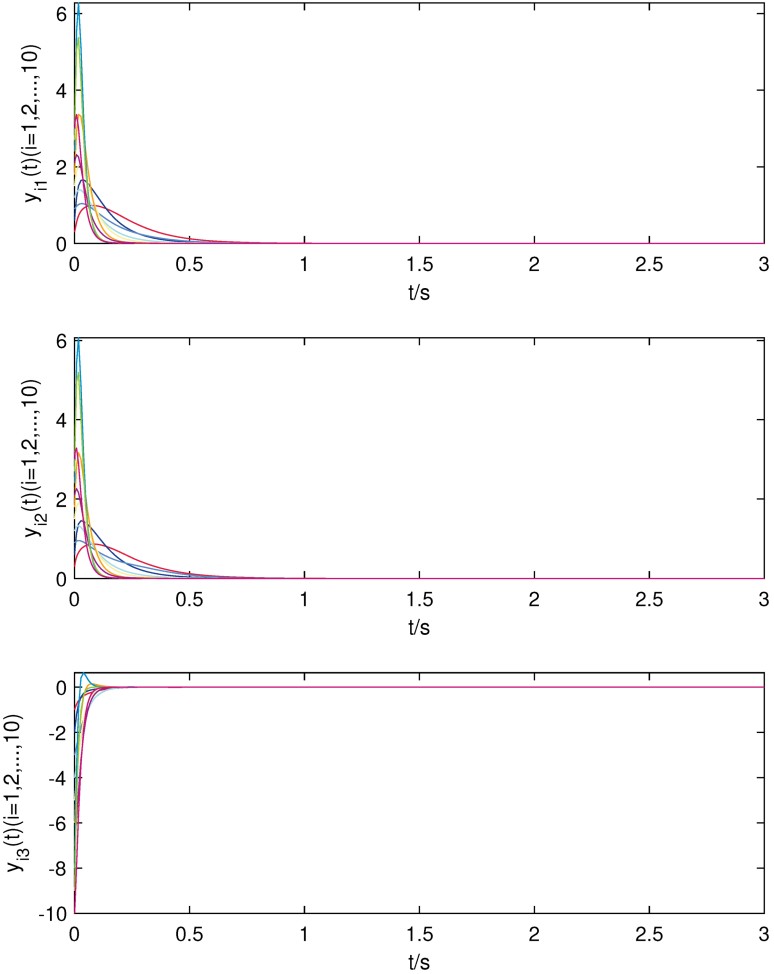

**Fig 11**. The state trajectories of nodes of sub-network Y in (25) with controller.

where $\beta_i = 0.08 + 0.02i$.

The desired formation vectors define the relative positions of each agent with respect to the formation center:

$$w_1 = \begin{bmatrix} 1 \\ 1 \end{bmatrix}, \quad w_2 = \begin{bmatrix} -1 \\ 1 \end{bmatrix}, \quad w_3 = \begin{bmatrix} -1 \\ -1 \end{bmatrix}, \quad w_4 = \begin{bmatrix} 1 \\ -1 \end{bmatrix} \tag{31}$$

$$r_1 = \begin{bmatrix} 1 \\ 1 \\ 0 \end{bmatrix}, \quad r_2 = \begin{bmatrix} -1 \\ 1 \\ 0 \end{bmatrix}, \quad r_3 = \begin{bmatrix} -1 \\ -1 \\ 0 \end{bmatrix}, \quad r_4 = \begin{bmatrix} 1 \\ -1 \\ 0 \end{bmatrix} \tag{32}$$

Correspondingly, the tracking errors for UGVs are defined as:

$$e_p^x = p^x - (p_d^x + w_i \tag{33}$$

$$e_v^x = v^x - 0 \tag{34}$$

The tracking errors for UAVs are defined as:

$$e_p^y = p^y - (p_d^y + r_i) \tag{35}$$

$$e_v^y = v^y - 0 \tag{36}$$

According to (8) and (9), the adaptive control laws are designed as:

$$u_i^x = -d_i^x(e_{pi}^x + e_{vi}^x) \tag{37}$$

$$u_i^y = -d_i^y(e_{pi}^y + e_{vi}^y) \tag{38}$$

To maintain the formation with UAVs maintaining a constant height of 2m above the corresponding UGVs, the coupling terms are:

$$H_x(x_i)x_j = [0_2, e_p^x + e_v^x]^T \tag{39}$$

$$H_y(y_i)y_j = [0_3, e_p^y + e_v^y]^T \tag{40}$$

$$H_{xy}(y_i)x_j = \begin{matrix} [0_3, -(p_{1i}^y - p_{1j}^x) - (v_{1i}^y - v_{1j}^x), -(p_{2i}^y - p_{2j}^x) - (v_{2i}^y - v_{2j}^x), \\ -(p_{3i}^y - 2) - v_{3i}^y]^T \end{matrix} \tag{41}$$

The initial conditions are set as: $x_1(0) = [0, 0, 0.1, 0.1]^T$, $x_2(0) = [3, 0, -0.1, 0.1]^T$, $x_3(0) = [6, 3, 0.1, -0.1]^T$, $x_4(0) = [3, 6, -0.1, -0.1]^T$ $y_1(0) = [1, 1, 3, 0.1, 0.1, 0]^T$, $y_2(0) = [4, 1, 4, -0.1, 0.1, 0]^T$, $y_3(0) = [7, 4, 5, 0.1, -0.1, 0]^T$, $y_4(0) = [4, 7, 6, -0.1, -0.1, 0]^T$ with initial adaptive gains $d_i^x(0) = [0.1, 0.12, 0.15, 0.13]$, $d_i^y(0) = [0.08, 0.1, 0.12, 0.09]$ and adaptive rates $k_i^x = [0.01, 0.012, 0.015, 0.011]$, $k_i^y = [0.008, 0.01, 0.012, 0.009]$.

Simulation results are depicted in Figs 12–15. Fig 12 shows the tracking errors of the UGVs, which converge asymptotically to zero. Fig 13 presents the tracking errors of the UAVs, which also vanish over time. Figs 14 and 15 illustrate the adaptive laws of the controller parameters for the UGVs and UAVs, respectively, where all adaptive parameters remain bounded.

In summary, the simulation results demonstrate that the proposed control law successfully achieves the UAV–UGV formation control objectives and exhibits practical applicability.

## Conclusion

In this paper, the stabilization of interdependent networks is achieved by decentralized adaptive controllers we proposed. In the model of the system, sub-networks neither need to have identical nodes nor the same number of nodes. Besides, the outer coupling matrices of them need not to satisfy the diffusive coupling condition, so it is a generalization of some models in previous literature. Furthermore, our controllers are applicable to interdependent networks with equally sized sub-networks. Finally, the effectiveness of the proposed controller is demonstrated by three simulation studies.

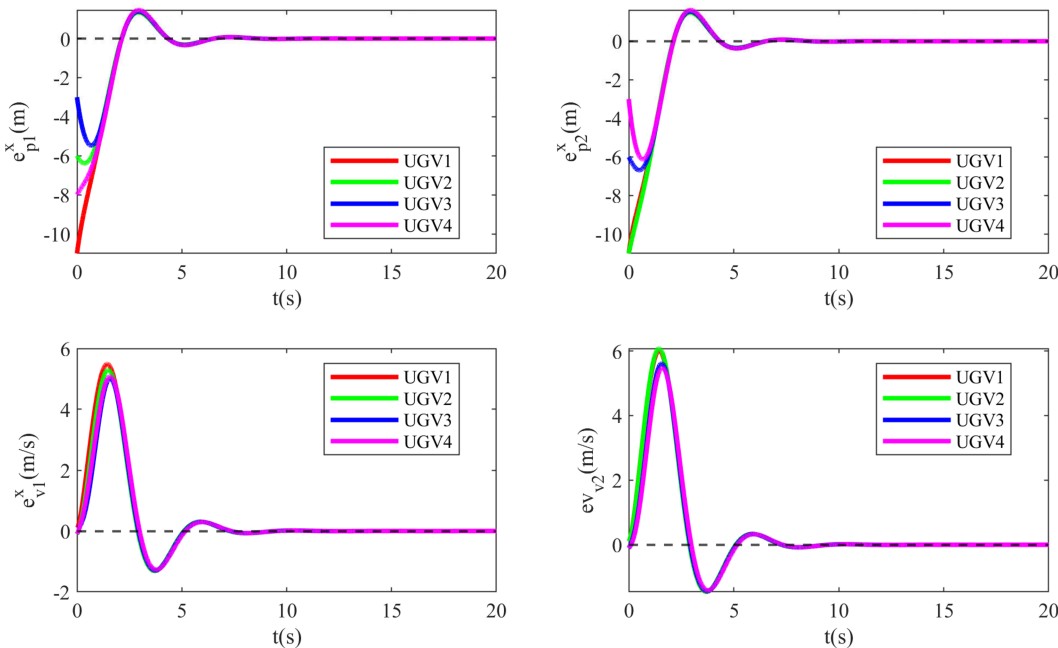

**Fig 12**. The tracking errors of the UGVs.

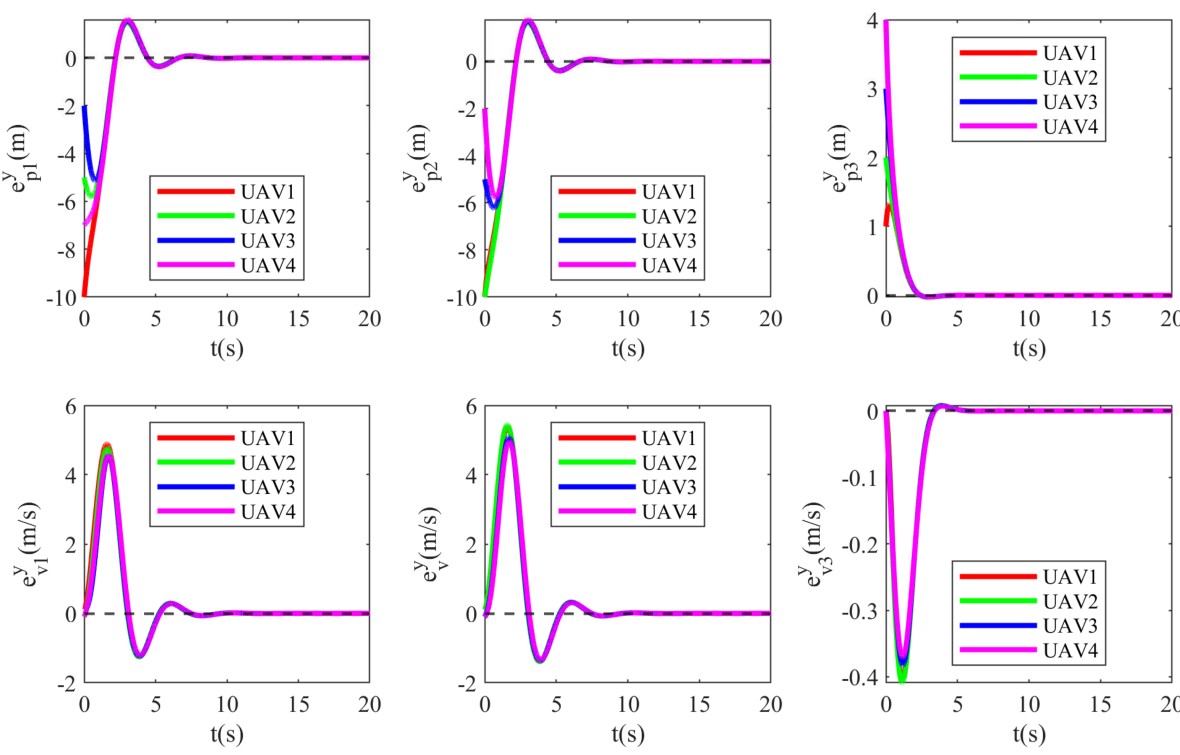

**Fig 13**. The tracking errors of the UAVs.

**Fig 14**. The evolution of the $d_x^i$ of UGVs.

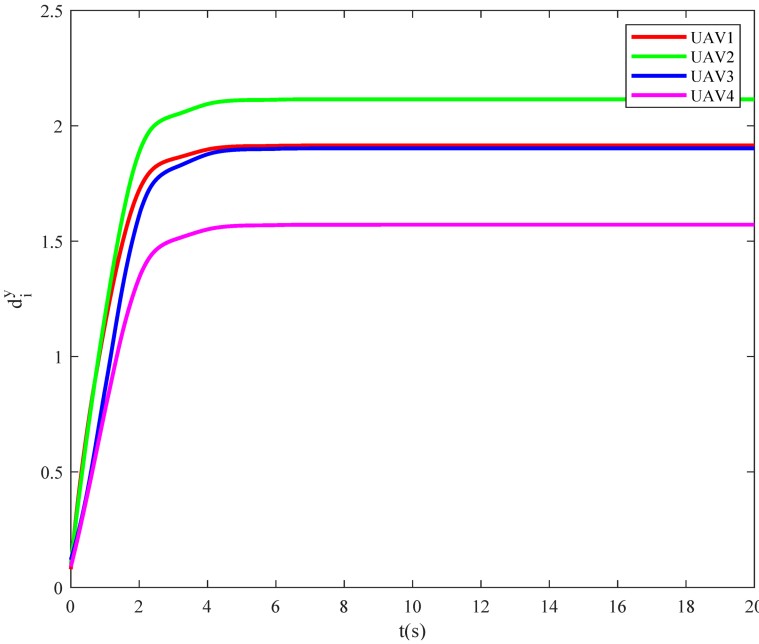

**Fig 15**. The evolution of the $d_y^i$ of UAVs.

## Supporting information

**S1 File. Matlab files of simulations. S1 Fig. A one-to-many interdependent network with two sub-networks *X* and *Y*.**
(TIFF)

**S2 Fig. The state trajectories of nodes of sub-network X in (22) without controller.**
(TIFF)

**S3 Fig. The state trajectories of nodes of sub-network Y in (23) without controller.**
(TIFF)

**S4 Fig. The state trajectories of nodes of sub-network X in (22) with controller.**
(TIFF)

**S5 Fig. The state trajectories of nodes of sub-network Y in (23) with controller.**
(TIFF)

**S6 Fig. The evolution of the $d_i^x(t)$ exerted on system (22)–(23).**
(TIFF)

**S7 Fig. The evolution of the $d_i^y(t)$ exerted on system (22)–(23).**
(TIFF)

**S8 Fig. The state trajectories of nodes of sub-network X in (24) without controller.**
(TIFF)

**S9 Fig. The state trajectories of nodes of sub-network Y in (25) without controller.**
(TIFF)

**S10 Fig. The state trajectories of nodes of sub-network X in (24) with controller.**
(TIFF)

**S11 Fig. The state trajectories of nodes of sub-network Y in (25) with controller.**
(TIFF)

**S12 Fig. The tracking errors of the UGVs.**
(TIFF)

**S13 Fig. The tracking errors of the UAVs.**
(TIFF)

**S14 Fig. The evolution of the $d_x^i$ of UGVs.**
(TIFF)

**S15 Fig. The evolution of the $d_y^i$ of UAVs.**
(TIFF)

## Author contributions

**Conceptualization:** Cao Chen, Changyuan Guo.

**Data curation:** Cao Chen, Tong Wang.

**Formal analysis:** Cao Chen.

**Funding acquisition:** Cao Chen, Changyuan Guo, Tong Wang.

**Investigation:** Cao Chen, Tong Wang.

**Methodology:** Cao Chen, Tong Wang.

**Project administration:** Changyuan Guo.

**Resources:** Changyuan Guo, Tong Wang.

**Software:** Cao Chen, Changyuan Guo, Tong Wang.

**Supervision:** Changyuan Guo, Tong Wang.

**Validation:** Changyuan Guo.

**Visualization:** Cao Chen, Tong Wang.

**Writing – original draft:** Cao Chen.

**Writing – review & editing:** Changyuan Guo.

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
