## [Decision Letter · Decision Letter 0]

20 Aug 2025

PONE-D-25-15972Stabilization of interdependent networks with two sub-networks of non-identical nodesPLOS ONE

Dear Dr.  Guo,

Thank you for submitting your manuscript to PLOS ONE. After careful consideration, we feel that it has merit but does not fully meet PLOS ONE’s publication criteria as it currently stands. Therefore, we invite you to submit a revised version of the manuscript that addresses the points raised during the review process.

We look forward to receiving your revised manuscript.

Kind regards,

Tiago Pereira

Academic Editor

PLOS ONE

Journal Requirements:

“Scientific and Technological Research Program of Chongqing Municipal Education Commission (KJQN202301218, Cao Chen ; KJQN202301219, Changyuan Guo and KJQN202301215, Tong Wang), Foundation of Intelligent Ecotourism Subject Group of Chongqing Three Gorges University (zhlv-20221024,Cao Chen ). These funds support us in preparation of the manuscript.”

Additional Editor Comments:

Please find mind and the referee report in the message below. My main concern is whether this paper has enough material to be considered an original contribution. Please elaborate on this carefully.

The central claim of the manuscript concerning stabilization of interdependent networks with non-identical nodes is essentially a straightforward extension of well-known control schemes. The "non-identical nodes" angle plays only a minor role: the proof just reuses standard Lipschitz-type assumptions and does not exploit any deeper structure. This makes the contribution marginal. The validation is also weak with toy numerical examples are shown. Both are deterministic toy models. There is no exploration of robustness, scalability, parameter sensitivity, or relevance to real interdependent networks. The paper claims importance for “transportation systems, power grids, communication networks,” but there is zero link to reality. This gap between the application claims and the toy theoretical exercise makes the paper feel disconnected and overstated.

Reviewer's Responses to Questions

**Comments to the Author**

1. Is the manuscript technically sound, and do the data support the conclusions?

Reviewer #1: Yes

2. Has the statistical analysis been performed appropriately and rigorously? 

Reviewer #1: N/A

3. Have the authors made all data underlying the findings in their manuscript fully available?

Reviewer #1: Yes

4. Is the manuscript presented in an intelligible fashion and written in standard English?

Reviewer #1: Yes

5. Review Comments to the Author

Reviewer #1: Report manuscript number: PONE-D-25-15972

As a whole the manuscript “Stabilization of interdependent networks with two sub-networks of non-identical nodes” is interesting and could be published on PLOS ONE, but befre that a few points need t be addressed:

In the introduction the authors insist on the importance of having sub-networks with different nodes.

But when they present their model, see eq. 1 and 2 are very similar. In particular the authors are not clear whether or not the functions f and g are equal or just represent the same (in this case dynamics) in the equations. This is crucial to the paper. Please explain.

Other points:

Assumption 1: what is N? Are both sub-networks equal?

Please, explain the meaning of assumption 2 (eq 3) and assumption 3 (eq 4).

6. PLOS authors have the option to publish the peer review history of their article (what does this mean?). If published, this will include your full peer review and any attached files.

Reviewer #1: No

---

## [Author Response · Author response to Decision Letter 1]

19 Sep 2025

Response to Reviewer Comments

Paper No. : PONE-D-25-15972

Title: Stabilization of interdependent networks with two sub-networks of non-identical nodes

Authors: Cao Chen, Changyuan Guo, Tong Wang

Response to Editor:

[General Comment]:

Thank you for submitting your manuscript to PLOS ONE. After careful consideration, we feel that it has merit but does not fully meet PLOS ONE’s publication criteria as it currently stands. Therefore, we invite you to submit a revised version of the manuscript that addresses the points raised during the review process.

[Reply]:

We would like to sincerely thank you for handling our manuscript entitled “Stabilization of interdependent networks with two sub-networks of non-identical nodes”. We appreciate your recognition that the manuscript has merit, and we fully understand your concern that, in its current form, it does not yet fully meet the publication criteria of PLOS ONE.

In the revised version, we have carefully addressed the issues raised during the review process. We believe that these revisions improve the manuscript and bring it in line with the publication criteria of PLOS ONE. We are sincerely grateful for your constructive feedback and for providing us with the opportunity to strengthen our work.

Due to the limitations of the submission system, the detailed point-by-point responses, including formulas, figures, and tables, cannot be clearly displayed in this text box. Therefore, we have provided a comprehensive Response to Reviewer document, which includes all detailed replies to the reviewers’ and editor’s comments. Please kindly refer to that file for the complete responses.

---

## [Decision Letter · Decision Letter 1]

4 Nov 2025

PONE-D-25-15972R1Stabilization of interdependent networks with two sub-networks of non-identical nodesPLOS ONE

Dear Dr. Guo,

Thank you for submitting your manuscript to PLOS ONE. After careful consideration, the referee still has suggestions to improve your manuscript. Therefore, we invite you to submit a revised version of the manuscript that addresses the points raised during the review process.

We look forward to receiving your revised manuscript.

Kind regards,

Tiago Pereira

Academic Editor

PLOS ONE

Journal Requirements:

Reviewers' comments:

Reviewer's Responses to Questions

**Comments to the Author**

1. If the authors have adequately addressed your comments raised in a previous round of review and you feel that this manuscript is now acceptable for publication, you may indicate that here to bypass the “Comments to the Author” section, enter your conflict of interest statement in the “Confidential to Editor” section, and submit your "Accept" recommendation.

Reviewer #1: (No Response)

2. Is the manuscript technically sound, and do the data support the conclusions?

Reviewer #1: Yes

3. Has the statistical analysis been performed appropriately and rigorously? 

Reviewer #1: Yes

4. Have the authors made all data underlying the findings in their manuscript fully available?

Reviewer #1: Yes

5. Is the manuscript presented in an intelligible fashion and written in standard English?

Reviewer #1: Yes

6. Review Comments to the Author

Reviewer #1: I am satisfied with the authros responses. I think that now the manuscript can be considered for publicatio in PLOSONE, after correcting a few minor problems, which I point below.

Introduction:

4th paragraph: Explain an acronym (QUAD) the first time it appears in the manuscript.

Preliminaries and model formulations:

Check Capital letters!

Example 3.

Before equation (26) the aauthors mention: the UAV–UGV group is composed of 4

UAVs and 4 UGVs, and their dynamic models are given as follows:

But later they say: Dxi ∈ R4×4 and Dyi ∈ R6×6, which is clear in eqs. 29 and 30.

Correct the phrase before Eq. 26!

7. PLOS authors have the option to publish the peer review history of their article (what does this mean?). If published, this will include your full peer review and any attached files.

Reviewer #1: No

---

## [Author Response · Author response to Decision Letter 2]

11 Nov 2025

Dear Editor and Reviewers,

Thank you very much for your time and for providing valuable comments and constructive suggestions on our manuscript. These comments have been instrumental in improving the quality of our work.

We have thoroughly revised the manuscript according to all the feedback received. The point-by-point response to all comments and suggestions is provided in the separate file named "Response to Reviewers.docx", which has been uploaded with the revised submission.

We appreciate your expertise and consideration.

Best regards,

Changyuan Guo

---

## [Editor Report · Decision Letter 2]

16 Nov 2025

Stabilization of interdependent networks with two sub-networks of non-identical nodes

PONE-D-25-15972R2

Dear Dr. Guo,

We’re pleased to inform you that your manuscript has been judged scientifically suitable for publication and will be formally accepted for publication once it meets all outstanding technical requirements.

Kind regards,

Tiago Pereira

Academic Editor

PLOS ONE
---

## [Editor Report · Acceptance letter]

PONE-D-25-15972R2

PLOS One

Dear Dr. Guo,

I'm pleased to inform you that your manuscript has been deemed suitable for publication in PLOS One. Congratulations! Your manuscript is now being handed over to our production team.

Kind regards,

on behalf of

Dr. Tiago Pereira

Academic Editor

PLOS One